# Rational Design, Synthesis and Preliminary Evaluation of Novel Fusarinine C-Based Chelators for Radiolabeling with Zirconium-89

**DOI:** 10.3390/biom9030091

**Published:** 2019-03-06

**Authors:** Chuangyan Zhai, Shanzhen He, Yunjie Ye, Christine Rangger, Piriya Kaeopookum, Dominik Summer, Hubertus Haas, Leopold Kremser, Herbert Lindner, Julie Foster, Jane Sosabowski, Clemens Decristoforo

**Affiliations:** 1School of Forensic Medicine, Southern Medical University, 510515 Guangzhou, China; zhaichuangyan@smu.edu.cn; 2Department of Nuclear Medicine, Medical University Innsbruck, 6020 Innsbruck, Austria; christine.rangger@i-med.ac.at (C.R.); gamsuk@hotmail.com (P.K.); Dominik.summer@i-med.ac.at (D.S.); 3Department of Nuclear Medicine, Guangdong Academy of Medical Sciences, 510080 Guangzhou, China; candido.he@163.com (S.H.); gdsy1970@163.com (Y.Y.); 4Division of Molecular Biology, Biocenter, Medical University Innsbruck, 6020 Innsbruck, Austria; Hubertus.Haas@i-med.ac.at; 5Division of Clinical Biochemistry, Biocenter, Medical University of Innsbruck, 6020 Innsbruck, Austria; Leopold.Kremser@i-med.ac.at (L.K.); Herbert.Lindner@i-med.ac.at (H.L.); 6Centre for Molecular Oncology, Barts Cancer Institute, Queen Mary University of London, E1 4NS London, UK; j.m.foster@qmul.ac.uk (J.F.); j.k.sosabowski@qmul.ac.uk (J.S.)

**Keywords:** fusarinine C (FSC), zirconium-89, bifunctional chelator, immuno-positron emission tomography (PET)

## Abstract

Fusarinine C (FSC) has recently been shown to be a promising and novel chelator for ^89^Zr. Here, FSC has been further derivatized to optimize the complexation properties of FSC-based chelators for ^89^Zr-labeling by introducing additional carboxylic groups. These were expected to improve the stability of ^89^Zr-complexes by saturating the 8-coordination sphere of [^89^Zr] Zr^4+^, and also to introduce functionalities suitable for conjugation to targeting vectors such as monoclonal antibodies. For proof of concept, succinic acid derivatization at the amine groups of FSC was carried out, resulting in FSC(succ)_2_ and FSC(succ)_3_. FSC(succ)_2_ was further derivatized to FSC(succ)_2_ AA by reacting with acetic anhydride (AA). The Zr^4+^ complexation properties of these chelators were studied by reacting with ZrCl_4_. Partition coefficient, protein binding, serum stability, acid dissociation, and transchelation studies of ^89^Zr-complexes were carried out in vitro and the results were compared with those for ^89^Zr-desferrioxamine B ([^89^Zr]Zr-DFO) and ^89^Zr-triacetylfusarinine C ([^89^Zr]Zr-TAFC). The in vivo properties of [^89^Zr]Zr-FSC(succ)_3_ were further compared with [^89^Zr]Zr-TAFC in BALB/c mice using micro-positron emission tomography/computer tomography (microPET/CT) imaging. Fusarinine C (succ)_2_AA and FSC(succ)_3_ were synthesized with satisfactory yields. Complexation with ZrCl_4_ was achieved using a simple strategy resulting in high-purity Zr-FSC(succ)_2_AA and Zr-FSC(succ)_3_ with 1:1 stoichiometry. Distribution coefficients of ^89^Zr-complexes revealed increased hydrophilic character compared to [^89^Zr]Zr-TAFC. All radioligands showed high stability in phosphate buffered saline (PBS) and human serum and low protein-bound activity over a period of seven days. Acid dissociation and transchelation studies exhibited a range of in vitro stabilities following the order: [^89^Zr]Zr-FSC(succ)_3_ > [^89^Zr]Zr-TAFC > [^89^Zr]Zr-FSC(succ)_2_AA >> [^89^Zr]Zr-DFO. Biodistribution studies of [^89^Zr]Zr-FSC(succ)_3_ revealed a slower excretion pattern compared to [^89^Zr]Zr-TAFC. In conclusion, [^89^Zr]Zr-FSC(succ)_3_ showed the best stability and inertness. The promising results obtained with [^89^Zr]Zr-FSC(succ)_2_AA highlight the potential of FSC(succ)_2_ as a monovalent chelator for conjugation to targeted biomolecules, in particular, monoclonal antibodies.

## 1. Introduction

Immuno-positron emission tomography (PET) is of great value for the development of monoclonal antibodies (mAbs) as therapeutic targeting vectors, enabling tracking and quantification of radiolabeled mAbs at high resolution and sensitivity [1]. In recent years, Zirconium-89 (^89^Zr), in particular, has become the most commonly-studied positron-emitting radionuclide for immuno-PET imaging. The physical half-life of 78.4 h is a good match with the biological half-life of mAbs, thus increasing the utility of PET in the development of this class of compound, and its medium mean positron energy of 0.395 MeV allows high-resolution PET imaging [2]. Additionally, in contrast to iodine-124, another long-lived positron-emitter, which has a rapid release of radioiodine from the cell after internalization, ^89^Zr is a residualizing radiometal and facilitates the visualization of tumors [3]. As an osteophilic radionuclide, immuno-PET imaging with ^89^Zr requires stable coordination of the radionuclide with the chelator to minimize dissociation resulting in bone accumulation in vivo.

To date, desferrioxamine B (DFO), see Figure 1, has been the most widely used bifunctional chelating system for ^89^Zr-based radiopharmaceuticals. Although chelation with DFO results in complexes with acceptable stability in pre-clinical as well as some clinical studies, it is far from being optimal due to the partial release of [^89^Zr]Zr^4+^ from the chelating system [4]. The released [^89^Zr]Zr^4+^ has been reported to accumulate in bone, particularly at late time points (3–7 days), which results in a decrease in image contrast, as well as an increase in radiation dose, especially to bone marrow [5,6,7,8,9,10]. The instability of DFO is attributed to the linear structure (which allows easy access to challenging cations and natural chelators in vivo), as well as to the incomplete saturation of the 8-coordination sphere of Zr^4+^ [11]. The development of novel high-stability Zr^4+^-ligands to minimize the uptake of liberated Zr^4+^ in the bone and other non-targeted tissue is an interesting and important goal. While many investigators focus on the optimization of the conjugation moiety between DFO and the biomolecule, attempts to develop new chelators are increasing [12,13,14,15,16,17,18,19,20,21,22]. Several novel octadentate and oxygen-rich ^89^Zr-chelators have been reported showing improved in vitro stability. These novel chelators are designed to be acyclic or macrocyclic by the addition of another hydroxamate unit [12,13,17,19,21] or introduce new chelating groups such as hydroxypyridinone and pyrocatechol groups [14,15,16,20].

Recently, we reported that fusarinine C (FSC), which has a 36-membered ring structure and three bidentate hydroxamates to coordinate ^89^Zr, showed superior stability and kinetic inertness compared to DFO. In particular, the excellent targeting properties of FSC bioconjugates in PET-imaging studies showed FSC to be a highly promising alternative chelator for ^89^Zr [23,24,25]. However, since FSC is hexadentate rather than octadentate, we postulated that improvements could be expected by introducing additional coordinating groups to saturate the coordination sphere of [^89^Zr]Zr^4+^. Another potential drawback of FSC as a ^89^Zr-bifunctional chelator lies in it having three functionalities for derivatization, which, despite facilitating the employment of multivalent concepts for small molecules, are not suitable for conjugation to macromolecules, especially antibodies. Therefore, a further aim of this study was to design a FSC-based mono-functional ^89^Zr-chelator. As proof of concept, modifications via succinic acid derivatization resulting in FSC(succ)_2_ acetic anhydride (AA) and FSC(succ)_3_, see Figure 1, were developed. This straightforward modification introduces additional carboxylic acid groups known to coordinate with Zr, which additionally can be utilized for bioconjugation strategies. In this study, the in vitro stability and transchelation properties of these ^89^Zr-complexes were investigated and compared with [^89^Zr]Zr-DFO and ^89^Zr-triacetylfusarinine C ([^89^Zr]Zr-TAFC). The in vivo properties of [^89^Zr]Zr-FSC(succ)_3_ were further compared with [^89^Zr]Zr-TAFC in BALB/c mice.

## 2. Materials and Methods

All commercially available reagents were of reagent grade and used as supplied with no further purification. Desferrioxamine B was obtained from Genaxxon Bioscience GmbH (Ulm, Germany). Triacetylfusarinine C was prepared and purified as described previously [23]. [^89^Zr]Zr-oxalate was purchased from Perkin Elmer, Inc (Waltham, MA, US) with an activity concentration of 900–1000 MBq per milliliter.

Purification of compounds was performed via a preparative reversed-phase high-performance liquid chromatography system (preparative RP-HPLC) using a Gilson 322 HPLC pump, a Gilson ultraviolet (UV)/VIS-155 detector (Gilson International B.V., Limburg, Germany), and a Eurosil Bioselect Vertex Plus 300 × 8 mm 5 μm C_18A_ 300 Å column (Knauer, Berlin, Germany) with a flow rate of 3.0 mL/min. The gradient was as follows: 0–1.0 min 10% acetonitrile (CH_3_CN), 1.0–18.0 min 10–40% CH_3_CN (gradient A).

Analytical RP-HPLC analysis was performed with an UltiMate 3000 RS HPLC pump, an UltiMate 3000 RS column compartment (column oven temperature was set at 25 °C), an UltiMate 3000 UV-Vis variable wavelength detector (Dionex, Gemering, Germany), and a Raytest radiometric detector (Raytest GmbH, Straubenhardt, Germany). A Vydac 218 TP5215, 150 × 3.0 mm column (SRD, Vienna, Austria), flow rate 1.0 mL/min, and UV-Vis detection at 220/410 nm were employed with the following CH_3_CN/H_2_O/0.1% trifluoroacetic acid (TFA) gradient: 0–0.5 min 0% CH_3_CN/0.1% TFA, 0.5–7.0 min, 0–55% CH_3_CN/0.1% TFA (gradient B).

### 2.1. Fusarinine C

Fusarinine C was produced as described previously [23]. Briefly, the *Aspergillus fumigatus* mutant strain *∆sidG* was cultured in iron-free minimal medium with 1% glucose for 36 h at 37 °C and 200 rpm. Biomass was removed by filtration. The media-containing siderophore was collected and concentrated. Additional purification was performed via preparative RP-HPLC (gradient A, t_R_ = 10.2 min) followed by concentration and lyophilization. Analytical RP-HPLC (gradient B): t_R_ = 4.7 min.

### 2.2. Fusarinine C (succ)_3_

To FSC (50 mg, 69 µmol) dissolved in 1 mL dimethylformamide (DMF), 34.5 mg of succinic anhydride (5 eq, 0.34 mmol) was added and the pH was adjusted to 4.5 using *N*,*N*-diisopropylethylamine (DIPEA). After shaking for 0.5 h, FSC(succ)_3_ was directly isolated via preparative RP-HPLC (gradient A, t_R_ = 12.1 min) and confirmed by matrix-assisted laser desorption/ionization time of flight mass spectrometry (MALDI-TOF-MS). The collected fractions were concentrated and lyophilized. FSC(succ)_3_: yield: 63 mg, (61 µmol), 88%; MALDI TOF-MS: [M + H]^+^ = 1028.2 [C_45_H_66_N_6_O_21_; exact mass: 1026.4 (calculated)]. Analytical RP-HPLC (gradient B): t_R_ = 5.6 min.

### 2.3. Fusarinine C(succ)_2_

The general synthesis procedure of FSC(succ)_2_ was the same as that of FSC(succ)_3_ except that a 2 eq molar excess of succinic anhydride was used. FSC(succ)_2_ was isolated from the byproducts (FSC-succ and FSC(succ)_3_) via preparative RP-HPLC (gradient A, t_R_ = 11.5 min). FSC(succ)_2_: yield: 28 mg, (30 µmol), 43%. MALDI TOF-MS: [M + H]^+^ = 928.2 [C_41_H_62_N_6_O_18_; exact mass: 926.4 (calculated)]. Analytical RP-HPLC (gradient B): t_R_ = 5.4 min.

### 2.4. Fusarinine C(succ)_2_AA

To FSC(succ)_2_ (10 mg, 11 µmol) dissolved in 0.5 mL DMF, 1 mL of acetic anhydride (AA) was added. After shaking for 10 min, FSC(succ)_2_AA was quickly isolated via preparative RP-HPLC (gradient A, t_R_ = 11.9 min) and confirmed by MALDI-TOF-MS. The collected fractions were concentrated and lyophilized. FSC(succ)_2_AA: yield: 9 mg, (9.3 µmol), 85%. MALDI TOF-MS: [M + H]^+^ = 970.1 [C_43_H_64_N_6_O_19_; exact mass: 968.4 (calculated)]. Analytical RP-HPLC (gradient B): t_R_ = 5.6 min.

### 2.5. Zr-Fusarinine C(succ)_2_AA and Zr-Fusarinine C(succ)_3_.

ZrCl_4_ (121 mg) was dissolved in 1 mL 0.1 M hydrochloric acid solution resulting in a concentration of 0.52 M. After dissolving FSC(succ)_2_AA (5 mg, 5.2 µmol) or FSC(succ)_3_ (5 mg, 4.9 µmol) in 1 mL water, a 1.2 eq molar excess of ZrCl_4_ (11.9 µL and 11.3 µL, respectively) was added. The reaction solutions were shaken gently for 5 min, then loaded onto a pre-activated Sep-Pak C_18_ cartridge. After washing the cartridge with 10 mL water, Zr-FSC(succ)_2_AA and Zr-FSC(succ)_3_ were eluted with 1mL methanol. The eluate fractions were dried by evaporation under nitrogen and weighed. The identities of the products were confirmed by electrospray ionization mass spectrometry (ESI-MS). Zr-FSC(succ)_2_AA: yield: 3.6 mg, (3.4 µmol), 65%. ESI-MS: [M + H]^+^ = 1,055.3 [C_43_H_60_N_6_O_19_Zr, see Figure 2A; exact mass: 1054.3 (calculated)]. RP-HPLC (gradient B): t_R_ = 5.4 min. Zr-FSC(succ)_3_: yield: 3.7 mg, (3.3 µmol), 67%. ESI-MS: [M + H]^+^ = 1113.3 [C_45_H_62_N_6_O_21_Zr, see Figure 2B; exact mass: 1112.3 (calculated)]. Analytical RP-HPLC (gradient B): t_R_ = 5.3 min.

### 2.6. ^89^Zr-Labeling

Approximately 30 MBq (30 µL) of [^89^Zr]Zr-oxalate was mixed with 27 μL of sodium carbonate (Na_2_CO_3_, 1 M) and incubated for 3 min at room temperature (RT) [26]. Thereafter, 100 μL of 4-(2-hydroxyethyl)-1-piperazineëthanesulfonic acid (HEPES) buffer (0.5 M, pH 7.0) was added to the reaction vial. Either DFO (32.8 μg), TAFC (42.6 μg), FSC(succ)_2_AA (48.5 μg) or FSC(succ)_3_ (51.4 μg) was added to the reaction vial. The labeling mixture was allowed to react at RT with a pH between 6.8 and 7.2 for 90 min. The reaction progress was monitored via analytical RP-HPLC (gradient B): [^89^Zr]Zr-DFO: t_R_ = 4.8 min, [^89^Zr]Zr-TAFC: t_R_ = 5.9 min, [^89^Zr]Zr-FSC(succ)_2_AA: t_R_ = 6.1 min, and [^89^Zr]Zr-FSC(succ)_3_: t_R_ = 6.0 min.

For animal experiments, 60 µL of CaCl_2_ (0.5 M) was added to the radiolabeling solution and a precipitate of Ca-oxalate appeared. The solution was then passed through a 0.2-µm filter to remove Ca-oxalate which may cause kidney failure. Subsequently, the filtrate was diluted to an appropriate volume using saline for administration to the BALB/c mice.

### 2.7. In Vitro Characterization

#### 2.7.1. Distribution Coefficient

A volume of 500 µL PBS including 0.5 MBq radioligand ([^89^Zr]Zr-FSC(succ)_2_AA or [^89^Zr]Zr-FSC(succ)_3_) was combined with 500 µL octanol. The mixture was vortexed for 15 min, and centrifuged for 2 min at 2000 rcf. Subsequently, 50 µL aliquots of the aqueous and the octanol layer were collected, measured in a 2480 Wizard^2^ Automatic Gamma Counter (Perkin Elmer, Vienna, Austria), and distribution coefficient (logD) values were calculated using Excel 2013 (*n* = 5) (Redmond, WA, USA).

#### 2.7.2. Stability Assay

Determination of the stability of [^89^Zr]Zr-DFO, [^89^Zr]Zr-TAFC, [^89^Zr]Zr-FSC(succ)_2_AA and [^89^Zr]Zr-FSC(succ)_3_ was carried out by incubating the radioligands in PBS, a 1000-fold molar excess of ethylenediaminetetraacetic acid (EDTA) solution (radioligand vs. EDTA: 25 µM vs. 25 mM) with different pH (pH 7, pH 6, and pH 4), as well as in human serum (the human serum was obtained from author C.D. with his consent at the Medical University Innsbruck within a routine blood withdrawal procedure) for seven days at 37 °C. At selected time points, PBS and EDTA samples were analyzed in triplicate directly via analytical RP-HPLC, while serum aliquots were mixed with 500 µL of CH_3_CN, vortexed, and centrifuged at 20,000 rcf for 2 min. Then, the precipitate was washed three times using CH_3_CN before analysis.

#### 2.7.3. Transchelation Study

[^89^Zr]Zr-DFO (50 µL) was mixed with a 1000-fold molar excess of either FSC(succ)_2_AA or FSC(succ)_3_ and [^89^Zr]Zr-TAFC (or [^89^Zr]Zr-FSC(succ)_2_AA, [^89^Zr]Zr-FSC(succ)_3_) (50 µL each) was mixed with a 1000-fold molar excess of DFO and incubated for seven days at RT. At selected time points, aliquots of the solutions were analyzed in triplicate directly via analytical RP-HPLC. The transchelation was determined by the ratio of [^89^Zr]Zr-DFO and [^89^Zr]Zr-FSC derivatives.

#### 2.7.4. Protein Binding Assay

The protein binding property was evaluated by incubating [^89^Zr]Zr-FSC(succ)_2_AA or [^89^Zr]Zr-FSC(succ)_3_ for seven days at 37 °C in fresh human serum. Subsequently, 30 µL of the solution was passed through a size exclusion spin column (MicroSpinG-50 column, GE Healthcare, Buckinghamshire, UK) via centrifugation at 2000 rcf for 2 min. Protein binding of the complexes in triplicate was determined by measuring the activity on the column (non-protein bound) and the activity in the eluate (protein bound) in the gamma counter.

#### 2.7.5. Biodistribution Study

All animal experiments were conducted in compliance with the Austrian animal protection laws and with the approval of the Austrian Ministry of Science (BMWFW-66.011/0049-WF/II/3b/2014) or were approved by the Animal Welfare and Ethical Review Body at Queen Mary University of London, and the UK Home Office (Project Licence 70/7603).

For the evaluation of biodistribution, four female BALB/c mice (Charles River Laboratories, Sulzfeld, Germany) were intravenously injected with [^89^Zr]Zr-FSC(succ)_3_ (1.5 MBq/mouse, 4.8 µg precursor) or [^89^Zr]Zr-TAFC (1.5 MBq/mouse, 4 µg) into the tail vein and sacrificed by cervical dislocation at 6 h post injection (p.i.). Organs (spleen, pancreas, stomach, intestine, kidney, liver, heart, and lung), blood, muscle tissue, and bone were dissected, weighed and then measured in the gamma counter to calculate the percentage of injected dose per gram of tissue (% ID/g). Statistical analysis was performed using an independent two-population *t*-test (significance level *p* = 0.05) and Origin 6.1 software (Northampton, MA, USA).

#### 2.7.6. Micro-Positron Emission Tomography/Computer Tomography Imaging

Micro-Positron Emission Tomography/computer tomography (MicroPET/CT) imaging studies were carried out using an Inveon microPET/CT scanner (Siemens Preclinical Solutions, Knoxville, TN, USA). Healthy female BALB/c mice were injected intravenously with 5 MBq of either [^89^Zr]Zr-FSC(succ)_3_ (16 µg, *n* = 1) or [^89^Zr]Zr-TAFC (13 µg, *n* = 1). MicroPET/CT images were acquired under general anesthesia (isoflurane/O_2_) for 20 min. Imaging data were recorded via static scans at 80 min and 24 h p.i. The microPET/CT scans were reconstructed and merged with OSEM3D-SPMAP (PET, matrix size 256 × 256) and Feldkamp (CT, Shepp Logan filter). For evaluation of bone uptake, image-based quantitation was carried out by selecting volumes of interest around each knee region and % ID/cm^3^ were calculated for both time points (VivoQuant image analysis software, Invicro LLC, Boston, MA, USA) and expressed as mean values of left and right knees to compare both compounds.

## 3. Results

### 3.1. Synthesis

The straightforward synthesis of FSC(succ)_2_AA and FSC(succ)_3_ is shown in Scheme 1. Fusarinine C was reacted with succinic anhydride at pH 4.5 resulting in FSC(succ)_3_ and FSC(succ)_2_ in satisfactory yields of 88 ± 4.0% and 43 ± 4.5%, respectively. The unreacted amine group of FSC(succ)_2_ was further derivatized with a molar excess of acetic anhydride resulting in the final compound FSC(succ)_2_AA. High-performance liquid chromatograms demonstrated the purity of compounds above 96%, see Appendix A. Zr-FSC(succ)_2_AA and Zr-FSC(succ)_3_ were produced by the complexation of precursors with a molar excess of ZrCl_4_ in 0.1 M HCl solution at pH 1 within 5 min. Reaction of excess of precursor with ZrCl_4_ (0.1 eq) showed a comparable reaction efficiency. ESI-MS demonstrated not only the metal-to-ligand ratio of 1:1 but also the high-purity of Zr-complexes by analyzing the distinctive isotopic ratios of zirconium, see Appendix A.

### 3.2. ^89^Zr-Labeling

The ^89^Zr-labeling procedure of DFO, TAFC, FSC(succ)_2_AA and FSC(succ)_3_ was as previously described [23]. Quantitative radiolabeling was achieved in HEPES buffer at RT between pH 6.8 to 7.2 within 90 min. Quality control was carried out by analytical RP-HPLC (instead of the instant thin-layer chromatography (ITLC) method for [^89^Zr]Zr-DFO and [^89^Zr]Zr-TAFC reported previously in which [^89^Zr]Zr-DFO and [^89^Zr]Zr-TAFC remain at the origin and free [^89^Zr]Zr^4+^ migrates with the solvent front). This change was due to the undesirable spread of [^89^Zr]Zr-FSC(succ)_2_AA and [^89^Zr]Zr-FSC(succ)_3_ along the ITLC strips.

### 3.3. Ex Vitro Characterization

Fusarinine C(succ)_2_AA and FSC(succ)_3_ showed high solubility in water, which is comparable to TAFC but different from recently reported novel octadentate chelators which seem to have poor solubility [9]. The lower logD values of [^89^Zr]Zr-FSC(succ)_2_AA and [^89^Zr]Zr-FSC(succ)_3_ (−3.3 ± 0.1 and −3.5 ± 0.4, respectively) indicate higher hydrophilic properties than [^89^Zr]Zr-DFO and [^89^Zr]Zr-TAFC (−3.0 ± 0.1 and −2.0 ± 0.0, respectively). All radioligands showed high stability in PBS and human serum over a period of seven days and no demetallation was observed. The low protein-bound activity (<10%) over the whole monitoring period further confirmed the stability of radioligands in human serum. The logD, protein binding and stability in human serum data for [^89^Zr]Zr-FSC(succ)_2_AA and [^89^Zr]Zr-FSC(succ)_3_ are summarized in Table 1 and compared with previously reported data of [^89^Zr]Zr-DFO and [^89^Zr]Zr-TAFC [23].

Acid dissociation experiments and transchelation studies were performed to compare the relative kinetic inertness of [^89^Zr]Zr-DFO, [^89^Zr]Zr-TAFC, [^89^Zr]Zr-FSC(succ)_2_AA and [^89^Zr]Zr-FSC(succ)_3_. Data of dissoctiation experiments are summarized in Figure 3. Briefly, at pH 7, [^89^Zr]Zr-TAFC, [^89^Zr]Zr-FSC(succ)_2_AA and [^89^Zr]Zr-FSC(succ)_3_ revealed excellent resistance to transchelation over seven days, in contrast to that for [^89^Zr]Zr-DFO where almost 60% transchelation occurred, see Figure 3A. At pH 6, [^89^Zr]Zr-FSC(succ)_3_ remained intact (97.8%), being more stable than [^89^Zr]Zr-TAFC (94.4%) and significantly higher than [^89^Zr]Zr-FSC(succ)_2_AA (76%), see Figure 3B. At pH 4, the difference in stability of [^89^Zr]Zr-FSC(succ)_3_ and [^89^Zr]Zr-TAFC was more pronounced (87.9 vs. 70.3% of intact complex, respectively, at day 7), see Figure 3C.

Transchelation experiments are shown in Figure 4 and were performed by incubating: (A) [^89^Zr]Zr-DFO in a 1000-fold molar excess of FSC(succ)_2_AA or FSC(succ)_3_, or (B) [^89^Zr]Zr-TAFC, [^89^Zr]Zr-FSC(succ)_2_AA and [^89^Zr]Zr-FSC(succ)_3_ incubated in a 1000-fold molar excess of DFO. The transchelation ratio of (A) ^89^Zr from [^89^Zr]Zr-DFO to FSC(succ)_2_AA and FSC(succ)_3_ or (B) from [^89^Zr]Zr-TAFC, [^89^Zr]Zr-FSC(succ)_2_AA and [^89^Zr]Zr-FSC(succ)_3_ to DFO was easily calculated due to the significant difference of the retention time of radioligands. Almost quantitative transchelation of ^89^Zr from [^89^Zr]Zr-DFO to FSC(succ)_2_AA or FSC(succ)_3_ was observed within 1 h. In contrast, [^89^Zr]Zr-TAFC, [^89^Zr]Zr-FSC(succ)_2_AA and [^89^Zr]Zr-FSC(succ)_3_ showed a stronger resistance to transchelation to DFO. After 7 days, [^89^Zr]Zr-FSC(succ)_3_ showed the highest kinetic inertness (62.1% of intact complex) compared with [^89^Zr]Zr-TAFC and [^89^Zr]Zr-FSC(succ)_2_AA (39.8% and 30.8% of intact complex, respectively).

### 3.4. Biodistribution Study and Micro-Positron Emission Tomography/Computer Tomography (microPET/CT) Imaging

Based on the above-mentioned results, [^89^Zr]Zr-FSC(succ)_3_ and [^89^Zr]Zr-TAFC were further compared in vivo. Biodistribution data of [^89^Zr]Zr-FSC(succ)_3_ and [^89^Zr]TAFC are presented in Figure 5. Both radioligands show rapid clearance from the body, mainly through the kidneys and a rather low uptake, especially, in bones. A relatively slower blood clearance for [^89^Zr]Zr-FSC(succ)_3_ than for [^89^Zr]Zr-TAFC was observed (0.11 ± 0.08 vs. 0.05 ± 0.01% ID/g). Correspondingly, activity in heart (0.28 ± 0.08% ID/g), lung (0.21 ± 0.04% ID/g), muscle (0.25 ± 0.07% ID/g), and bone (0.24 ± 0.04% ID/g) were also higher. The microPET/CT images of the BALB/c mouse at 80 min p.i. and 24 h p.i. injected with either [^89^Zr]Zr-FSC(succ)_3_ or [^89^Zr]Zr-TAFC further confirmed similar rapid pharmacokinetics, see Figure 6, three-dimensional images: Appendix A. The kidneys and bladder were the primarily visible structures, which is related to excretion and kidney retention. At 80 min p.i. [^89^Zr]Zr-FSC(succ)_3_ showed some gall bladder activity and some minor activity in the small bowel which disappeared in the late images. More importantly, no bone uptake was observed, confirming the high in vivo stability of both compounds. An image-based quantitative comparison of bone uptake in the knee of mice revealed an even lower uptake of [^89^Zr]Zr-FSC(succ)_3_ with 0.083 and 0.026% ID/cm^3^ and of [^89^Zr]Zr-TAFC with 0.040 and 0.016% ID/cm^3^ at 80 min and 24 h, respectively, underlining the high in vivo stability of both chelates.

## 4. Discussion

The high potential of FSC as a novel multivalent ^89^Zr-bifunctional chelator was demonstrated by our group recently [23]. However, the six coordinating oxygens of FSC do not coordinatively saturate the Zr^4+^. In addition, three of the functionalities of FSC may limit its application to the macromolecules, especially antibodies. On this basis, we were interested in designing novel chelators based on FSC by introducing two or three carboxylic functional groups, which were expected to, on one hand, improve the stability of complexation by saturating the coordination sphere of [^89^Zr]Zr^4+^ and, on the other hand, reduce the number of conjugation groups making them suitable for conjugation to mAbs.

As proof of concept, two or three succinic acid groups were introduced into the FSC skeleton and FSC(succ)_2_, FSC(succ)_2_AA and FSC(succ)_3_ were synthesized with satisfactory yields. In our previous report, we discovered that iron protection is required for coupling FSC with targeting vectors [23]. However, the present study revealed that FSC can couple active groups such as anhydride groups at acidic pH, which provides a simpler conjugation approach. Moreover, the successful synthesis of FSC(succ)_2_ and FSC(succ)_2_AA opens a way to apply it to mAbs. FSC(succ)_2_, which has only one free amine functionality left, possesses the potential to couple mAbs with an activated group resulting in a monomeric FSC(succ)_2_-mAb.

In this study, the rapid and quantitative complexing of FSC(succ)_2_AA and FSC(succ)_3_ with ZrCl_4_ was reported and confirmed by ESI-MS. In our previous papers, we reported simple quantitative binding of ZrCl_4_ with TAFC at acidic pH (1–5) within 5 min [23] and quantitative labeling of TAFC with ^89^Zr-oxalate using acetate buffer [27]. On the basis of these findings, we can anticipate that [^89^Zr]ZrCl_4_ may be a better choice for labeling FSC derivatives in acetate buffer at appropriate pH. The advantage of [^89^Zr]ZrCl_4_ over ^89^Zr-oxalate lies not only in the shorter labeling time but also in avoiding the use of highly concentrated oxalate solution which potentially can result in the production of solid calcium oxalate causing kidney damage. In contrast to HEPES buffer, which requires additional quality control before clinical application, acetate buffer is entirely pharmaceutically compatible. For better understanding the exact coordination structures as well as their influence on stability, the preparation of single crystals of Zr-TAFC, Zr-FSC(succ)_2_AA and Zr-FSC(succ)_3_ for X-ray Diffraction was explored using different strategies; however, this failed. Multivalent hydroxamates seem not to crystallize easily, and this may be the reason that Zr(Me-AHA)_4_ is the only successfully crystallized Zr-hydroxamate compound [11].

To assess the relative kinetic inertness of [^89^Zr]Zr-DFO, [^89^Zr]Zr-TAFC, [^89^Zr]Zr-FSC(succ)_2_AA and [^89^Zr]Zr-FSC(succ)_3_, acid dissociation experiments were performed under acidic conditions. At pH 4, [^89^Zr]Zr-FSC(succ)_3_ was the most kinetically inert compound to acid dissociation for seven days, with [^89^Zr]Zr-TAFC showing moderate stability, followed by [^89^Zr]Zr-FSC(succ)_2_AA, and [^89^Zr]Zr-DFO totally decomposing on day 3. Transchelation studies showed the same but more pronounced trend. [^89^Zr]Zr-DFO showed total transchelation within 1 h whereas other ^89^Zr-complexes only showed limited transchelation after seven days. The improved stability and kinetic inertness of [^89^Zr]Zr-FSC(succ)_3_, [^89^Zr]Zr-TAFC, and [^89^Zr]Zr-FSC(succ)_2_AA over [^89^Zr]Zr-DFO were attributed to the macrocyclic structure. The superior stability of [^89^Zr]Zr-FSC(succ)_3_ over [^89^Zr]Zr-TAFC and [^89^Zr]Zr-FSC(succ)_2_AA may be as a result of the higher coordination number and better coordination configuration. The lower stability of [^89^Zr]-Zr-FSC(succ)_2_AA compared to [^89^Zr]Zr-TAFC may be related to the loss of the symmetrical structure. However, the superior stability of [^89^Zr]Zr-FSC(succ)_2_AA over [^89^Zr]Zr-DFO highlights the potential of FSC(succ)_2_ as a monomeric chelator in the development of novel ^89^Zr-tracers.

A direct biodistribution comparison revealed a much slower blood clearance of [^89^Zr]Zr-FSC(succ)_3_ than [^89^Zr]Zr-TAFC, resulting in a higher uptake in all major organs. Notably, the bone activity of [^89^Zr]Zr-FSC(succ)_3_ was significantly higher than that of [^89^Zr]Zr-TAFC, possibly mainly reflecting the slower clearance rather than being actual bone uptake. Even though different time points were chosen for an imaging study, this was confirmed by images at 24 h p.i. showing both radioligands without visible bone uptake. [^89^Zr]Zr-FSC(succ)_3_ also showed some hepatobiliary excretion in microPET/CT images at the early time point of 80 min, which was not depicted at 24 h p.i. and in the 6 h p.i. biodistribution study. The overall difference in biodistribution may be attributed to the difference in the chelate-radiometal net charge and polarity. [^89^Zr]Zr-TAFC possesses a positive net charge while [^89^Zr]Zr-FSC(succ)_3_ shows an overall negative charge character, which may potentially influence the excretion patterns of radiotracers. However, the difference in clearance rates between [^89^Zr]Zr-FSC(succ)_3_ and [^89^Zr]Zr-TAFC should not be a concern in its final application. Upon conjugation to a targeting biomolecule, the pharmacokinetics of the [^89^Zr]Zr-FSC(succ)_3_ complex will be completely superseded by those of biomacromolecules. Higher kidney and lower liver uptake compared to [^89^Zr]Zr-TAFC indicates a renal excretion pathway, which is consistent with the higher hydrophilicity of [^89^Zr]Zr-FSC(succ)_3_.

## 5. Conclusions

In our study, three FSC derivatives were synthesized and labeled with zirconium-89. [^89^Zr]Zr-FSC(succ)_2_AA showed superiority over [^89^Zr]Zr-DFO in vitro. Furthermore, [^89^Zr]Zr-FSC(succ)_3_ revealed the best stability and inertness in comparison to [^89^Zr]Zr-DFO, [^89^Zr]Zr-TAFC and [^89^Zr]Zr-FSC(succ)_2_AA. The superiority of FSC(succ)_3_ was attributed to the higher coordination number and better coordination configuration. The successful synthesis of FSC(succ)_2_ and FSC(succ)_2_AA highlights the potential of FSC(succ)_2_ as a monovalent ^89^Zr-chelator for the conjugation to mAbs. Further studies of FSC(succ)_2_-conjugated mAbs, especially in vivo, in comparison with their DFO counterparts and potentially other novel Zr-chelators, are needed to show the true potential of this approach for Immuno-PET.

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
