# Peer review of "Rational Design, Synthesis and Preliminary Evaluation of Novel Fusarinine C-Based Chelators for Radiolabeling with Zirconium-89"

_biomolecules, 2019, doi:10.3390/biom9030091_

Reviewer 1 Report

The authors present a study on the synthesis and initial evaluation of several derivatives of Fusaranine C as novel chelators for 89Zr. The synthesis, in vitro characterization, and initial in vivo characterization is described. Overall, the data is well presented, the conclusions are supported by the data, and it will be of interest to those developing 89Zr labeled PET radiopharmaceuticals.

A few points to revise:

The intro reads as a "wall of text". Could be broken down into two paragraphs.

The micro PET images at 80 minutes for the succ3 conjugate demonstrate additional foci in the abdomen in addition to the kidneys and bladder. I suspect these are the gallbladder and maybe small bowel? Would include additional information in the results and discussion section to document the accurate biodistribution. Also, these don't correlate well with the bio distribution data, possibly due to differences in time points (80 min/24h vs. 6 hours?) Should include in discussion as well.

The authors present MS and HPLC data to confirm identity but nothing to demonstrate purity of the compounds. HPLCs of the purified compounds should be included in supplemental and either NMR or elemental analysis also presented, since these are new compounds.

In order to support the assertion that there is no bone uptake on the micro PET the authors should perform a region of interest analysis of the bones at both time points. 

The authors should compare their results against other 89Zr chelates published in the literature. Although these authors cite that DFO has problems with loss of 89Zr, I suspect that the same experiments performed with DFO as in this paper would also show no bone uptake as in this prior paper. https://www.sciencedirect.com/science/article/pii/S0969805110005214?via%3Dihub

Author Response

The intro reads as a "wall of text". Could be broken down into two paragraphs.

For more easy reading we divided the introduction part into three paragraphs.

The micro PET images at 80 minutes for the succ3 conjugate demonstrate additional foci in the abdomen in addition to the kidneys and bladder. I suspect these are the gallbladder and maybe small bowel? Would include additional information in the results and discussion section to document the accurate biodistribution.

We carefully re-evaluated the images and agree with this interpretation and have added a clear description to the results and discussion section. Additionally we now include the 3D images as supplementary information where this can be viewed in more detail.

Also, these don't correlate well with the bio distribution data, possibly due to differences in time points (80 min/24h vs. 6 hours?) Should include in discussion as well.

We have included a remark related to these differences in the discussion to address this more clearly.

The authors present MS and HPLC data to confirm identity but nothing to demonstrate purity of the compounds. HPLCs of the purified compounds should be included in supplemental and either NMR or elemental analysis also presented, since these are new compounds.

HPLC chromatograms of the purified compounds are now included into supplementary materials following your suggestion. The gradient used was more gently leading significant differences of retention time between FSC(succ)2, FSC(succ)2AA and FSC(succ)3. NMR or elemental analysis unfortunately are not available.

In order to support the assertion that there is no bone uptake on the micro PET the authors should perform a region of interest analysis of the bones at both time points.

We now have performed ROI analysis and included it in the manuscript. It clearly confirms the very low bone accumulation of the compounds

The authors should compare their results against other 89Zr chelates published in the literature. Although these authors cite that DFO has problems with loss of 89Zr, I suspect that the same experiments performed with DFO as in this paper would also show no bone uptake as in this prior paper. https://www.sciencedirect.com/science/article/pii/S0969805110005214?via%3Dihub

We agree that 89Zr-DFO would probably show a similar picture. It is not easy to make a conclusion only by the comparison of 89Zr-chelators themselves in vivo due to the rapid excretion of small chelators. In our study, we set harsh conditions (e.g. higher acidic solution, higher concentration of competitors) to test the stability of our chelators and compared with [89Zr]Zr-DFO and demonstrated the superiority of our chelators to DFO.

However, the most convincing evidence of the in vivo stability of 89Zr-chelators at the end should be provided by evaluating the 89Zr-chelators at late time point (e.g. 3-7 days) by conjugation to an antibody. The in vivo instability issues of [89Zr]Zr-DFO-antibodies were only observed at very late time point. But this kind of study would be influenced by many factors such as antibody, chelating system than complex stability. Therefore, in this initial study we only focused on the comparison of the in vitro stability of 89Zr-chelators and left the in vivo comparison as the future work, which we now more clearly addressed in the discussion.

Reviewer 2 Report

1.- In the results section,  authors should include the SD of the yield values of the synthesis since it looks that the reaction was performed just once. The authors should repeat the reaction at least twice more

2.- The physicochemical characterization of the new compounds its poor, it should be more complete, including other techniques like NMR-H or HPLC to confirm the purity of the sample.

3.- For the MicroPET/CT authors should increase the number of animals since N=1 is a very low value for the statistics. In addition, it is not clear if n=1 is per tracer or just one animal was used

4.- The draft should include all the RP-HPLC chromatograms to confirm the purity and data.

Author Response

Comments and Suggestions for Authors

1. In the results section, authors should include the SD of the yield values of the synthesis since it looks that the reaction was performed just once. The authors should repeat the reaction at least twice more

Several batches of FSC(succ)2AA and FSC(succ)3 were synthesized and used for labeling, preparation of cold Zr compounds, recrystallization and so on. We have now included the SD of the yield values of the synthesis following your suggestion.

2. The physicochemical characterization of the new compounds its poor, it should be more complete, including other techniques like NMR-H or HPLC to confirm the purity of the sample.

HPLC chromatograms of the purified compounds are included into supplementary materials following your suggestion. The gradient used was more gently leading significant differences of retention time between FSC(succ)2, FSC(succ)2AA and FSC(succ)3. Also the purity of the samples have been added into manuscript: HPLC chromatograms demonstrated the purity of compounds above 96% (Supp. FS1, FS2, FS3). NMR-H unfortunately is not available.

3. For the MicroPET/CT authors should increase the number of animals since N=1 is a very low value for the statistics. In addition, it is not clear if n=1 is per tracer or just one animal was used

We agree that n=1 is poor statistics. However the intention of our imaging experiments were to get a preliminary information about early and late pharmacokinetics of the ligands, not intended as a full in vivo characterization. We feel that a solid comparison should include conjugation of the chelators to an antibody with detailed information of the differences observed between our chelators and DFO. This, however, was out of scope of this study emphasizing on the initial synthesis, stability and radiolabeling results. We have addressed this in the discussion and also made it more clear how many animals were used.

4. The draft should include all the RP-HPLC chromatograms to confirm the purity and data.

We uploaded HPLC chromatograms of the compounds in supplementary materials following your suggestion.

Reviewer 3 Report

Dear Authors,

I read caryfully your paper, but I do not recommend it, in the present form, for the publication in Biomelcules Journal. Befor subbmiting your paper to another journal I advice you the following changes:

1.     Introduction: 

-explain why did you choose carboxylic groups (succ) as additional Zr-binding sites and not other cooridnation groups;

2.     Materials and methods

- Zr-FSC(succ)2 and Zr-FSC(succ)3: describe and explain better part: "After shaking (...) by ESI-MS”.

- Zr89-labeling: explain why CaCl2 was added to the animal samples.

3.     Biodistribution study: the experimental group (n=4) is to small; moreover, the level of data uncertainity was not calculated. Describe more the kind of BALB/c mice and why only female

4.     MicroPET/CT imaginig: the experimental group (n=1) is to small.

5.     Ex vitro characterization: Figure 3. The description together with the figure is unclear and do not fit to the section under this figure. Describe it better and separete the stability data from the competition studies.

6.     Biodistribution data: Figure 4. why did you collect data after 6h? this data DO NOT FIT to MicroPET/CT data after 80 min and 24h. Moreover, sign the organs in the Figure 5.

7.     Discussion. First part of this chapter should be moved to the introduction. Moreover, you mention effective hydroxypyrone derivative [your ref 15], which is more effective Zr-chelator, but you do not comapre your data with literature data.

Best regards

Author Response

Comments and Suggestions for Authors

Dear Authors,

I read carefully your paper, but I do not recommend it, in the present form, for the publication in Biomelcules Journal. Before submitting your paper to another journal I advice you the following changes:

1. Introduction:

-explain why did you choose carboxylic groups (succ) as additional Zr-binding sites and not other cooridnation groups;

We chose carboxylic groups due to ease of access of appropriate building blocks, introduction of additional hydroxamates would have been very challenging and we thought introduction of aromatic coordinations would have impaired biological properties introducing lipophilicity. Also it is known that [89Zr]Zr4+ coordinates with carboxylic groups, e.g. in DOTA-complexes. We have explained this rational in the introduction more clearly now

2. Materials and methods

- Zr-FSC(succ)2 and Zr-FSC(succ)3: describe and explain better part: "After shaking (...) by ESI-MS”.

Following your advice, we improved our description as followed: The reaction solutions was shaken gently by a shaker machine for 5 min, then loaded to a pre-activated Sep-Pak C18 cartridge. Zr-FSC(succ)2AA and Zr-FSC(succ)3 were eluted with 1 mL methanol after a procedure of 10 mL water washing cartridge. The eluate fractions were dried by a nitrogen evaporator and weighted. The identities of products were confirmed by ESI-MS.

- Zr89-labeling: explain why CaCl2 was added to the animal samples.

We initially were (maybe unnecessarily) reluctant to inject oxalic acid into the animals, which may cause kidney damage due to Ca-Oxalate precipitation. We have explained it following your suggestion also in the text.

3. Biodistribution study: the experimental group (n=4) is too small; moreover, the level of data uncertainty was not calculated. Describe more the kind of BALB/c mice and why only female

We agree that the data from more mice in a group is more convincing, but today it is becoming challenging to use more animals. We designed our experiment carefully and calculate the number of animal used. We think n=4 is statistically significant and, as an initial evaluation, is enough to explore the in vivo properties and biodistribution of our chelators. Female BALB/c mice (healthy or tumour model) are widely used in radiopharmacutical research and we also used this kind of mice in our previous research. This (and not any other type or species) type of mice were covered by the Austrian animal license covering the biodistribution experiments.

4. MicroPET/CT imaginig: the experimental group (n=1) is too small.

We agree that n=1 is poor statistics. However the intention of our imaging experiments were to get a preliminary information about early and late pharmacokinetics of the ligands, not intended as a full in vivo characterization. We feel that a solid comparison should include conjugation of the chelators to an antibody with detailed information of the differences observed between our chelators and DFO. This, however, was out of scope of this study emphasizing on the initial synthesis, stability and radiolabeling results. We have addressed this in the discussion and made it clear how many animals were used.

5. Ex vitro characterization: Figure 3. The description together with the figure is unclear and do not fit to the section under this figure. Describe it better and separate the stability data from the competition studies.

We have separated Figure 3D and made it a separate Figure 4 for clarification. Also the titles of Figure 3 and Figure 4 were optimized so they can be understood more easily.

6. Biodistribution data: Figure 4. why did you collect data after 6h? this data DO NOT FIT to MicroPET/CT data after 80 min and 24h.

We understand the argument that the time point of biodistrubution didn't match the imaging time points (80 min and 24 h). An early time point (e.g. 80 min) would facilitate to evaluate the in vivo properties of chelators themselves due to the rapid clearance of small chelators from the body, while late time points would give more information on the stability of 89Zr-chelators in vivo. That is why micro PET/CT imaging was performed at the 80 min and 24 h p.i. by our British colleagues whose lab is allowed to perform a several day-lasting animal experiment. The time point of biodistribution (6 h) was chosen based on the reality the animal license in Austria that did not allow us to do an overnight animal experiment. To avoid unnecessary animal testing and to address the potential differences between the two ligands we chose this experiments setup.

Moreover, sign the organs in the Figure 5.

We have signed the organs following your suggestion.

7. Discussion. First part of this chapter should be moved to the introduction.

We rewrote the introduction and discussion parts to make them more clear and concise following your suggestion. 

Moreover, you mention effective hydroxypyrone derivative [your ref 15], which is more effective Zr-chelator, but you do not comapre your data with literature data.

We think it would be quite confusing to compare our data with all potential 89Zr-chelators, therefore we concentrated on the “standard” DFO, however we now have introduced a remark to this in the final conclusion

Round  2

Reviewer 1 Report

I think concerns have been addressed. Recommend acceptance.

Author Response

Thank you 

Reviewer 3 Report

Dear Authors,

Thank you for your revision. Nevertheless, three point still remain unresolved:

3. Biodistribution study: the experimental group (n=4) is too small; moreover, the level of data uncertainty was not calculated. Describe more the kind of BALB/c mice and why only female

We agree that the data from more mice in a group is more convincing, but today it is becoming challenging to use more animals. 

I agree, for this reason the experiments must be planned careful. Otherwise 4 mice were sacrifice for nothing. 

We designed our experiment carefully and calculate the number of animal used. We think n=4 is statistically significant and, as an initial evaluation, is enough to explore the in vivo properties and biodistribution of our chelators. Female BALB/c mice (healthy or tumour model) are widely used in radiopharmacutical research and we also used this kind of mice in our previous research. This (and not any other type or species) type of mice were covered by the Austrian animal license covering the biodistribution experiments.

The level of data uncertainty (for n=4) is still not calculated.

4. MicroPET/CT imaginig: the experimental group (n=1) is too small.

We agree that n=1 is poor statistics. However the intention of our imaging experiments were to get a preliminary information about early and late pharmacokinetics of the ligands, not intended as a full in vivo characterization. 

At this point you should considered your paper a “short communication” not “full paper”. 

We feel that a solid comparison should include conjugation of the chelators to an antibody with detailed information of the differences observed between our chelators and DFO. This, however, was out of scope of this study emphasizing on the initial synthesis, stability and radiolabeling results. We have addressed this in the discussion and made it clear how many animals were used

6. Biodistribution data: Figure 4. why did you collect data after 6h? this data DO NOT FIT to MicroPET/CT data after 80 min and 24h.

We understand the argument that the time point of biodistrubution didn't match the imaging time points (80 min and 24 h). An early time point (e.g. 80 min) would facilitate to evaluate the in vivo properties of chelators themselves due to the rapid clearance of small chelators from the body, while late time points would give more information on the stability of 89Zr-chelators in vivo. That is why micro PET/CT imaging was performed at the 80 min and 24 h p.i. by our British colleagues whose lab is allowed to perform a several day-lasting animal experiment. The time point of biodistribution (6 h) was chosen based on the reality the animal license in Austria that did not allow us to do an overnight animal experiment

It is very unclear explanation and the fact that results from different experiments do not fit still remains unresolved. 

To avoid unnecessary animal testing and to address the potential differences between the two ligands we chose this experiments setup.

Author Response

Dear Authors,

Thank you for your revision. Nevertheless, three point still remain unresolved:

3. Biodistribution study: the experimental group (n=4) is too small; moreover, the level of data uncertainty was not calculated. Describe more the kind of BALB/c mice and why only female

 The level of data uncertainty (for n=4) is still not calculated.

Regarding the number of animals, we are in line with most recent publications where animal biodistribution is described (e.g. Schottelius M, et al. Synthesis and Preclinical Characterization of the PSMA-Targeted Hybrid Tracer PSMA-I&F for Nuclear and Fluorescence Imaging of Prostate Cancer [J]. Journal of Nuclear Medicine, 2019, 60(1): 71-78. ).

We believe that the number of animals is sufficient to obtain information about the major differences between our 2 compounds studied. We do not think that the differences we found are a major concern, considering the final application when attached to an antibody.

However to address this more clearly we now included also the statistical comparison of the two groups in Fig. 5

4. MicroPET/CT imaginig: the experimental group (n=1) is too small.

At this point you should considered your paper a “short communication” not “full paper”.

We agree that the paper can be considered as a short communication and we have no objections to that, in particular as we see the major value in the synthesis and in vitro evaluation (stability evaluation, etc). The in vivo data were done to complement and support these major findings. However, we did not find any formatting requirements. We now have shortened and simplified some statements to address this issue and have communicated this to the editorial office

6. Biodistribution data: Figure 4. why did you collect data after 6h? this data DO NOT FIT to MicroPET/CT data after 80 min and 24h.

It is very unclear explanation and the fact that results from different experiments do not fit still remains unresolved

We do not think that the MicroPET/CT do not fit the biodistribution data. It should be seen that the values in the biodistribution data of remaining activity 6h p.i. are very low (mostly<1%ID/g), therefore it is expected that some organs are not visible in PET/CT. The higher intestinal activity and visualization in gall bladder of [89Zr]Zr-FSC(succ)3 is not reflected in the biodistribution study, however, it can be expected that activity has cleared after 6hr, which was also the case in the 24hr PET/CT image. One could argue that the PET/CT image 24hr delineates only kidneys and seems to be higher than in the biodistribution study, however this impression is misleading considering the low activity levels and also that biodistribution reflects %ID/g whereas images are “vizualisation”, which is also dependent on organ size and localization of the organ and the average level of activity throughout the organ is high enough to be above background activity.